# Rats spontaneously perceive global motion direction of drifting plaids

**Giulio Matteucci**[¤a©], **Benedetta Zattera**[¤b©], **Rosilari Bellacosa Marotti**,
**Davide Zoccolan** *

Visual Neuroscience Lab, International School for Advanced Studies (SISSA), Trieste, Italy

© These authors contributed equally to this work.
¤a Current address: Department of Basic Neurosciences, University of Geneva, Geneva, Switzerland
¤b Current address: Friedrich Miescher Institute for Biomedical Research (FMI), Basel, Switzerland
* zoccolan@sissa.it

**Data Availability Statement:** The authors confirm that all data and code underlying the findings of this study are fully available without restriction. The experimental raw data analyzed and the code underlying the simulations are available on Zenodo under accession code

## Abstract

Computing global motion direction of extended visual objects is a hallmark of primate high-level vision. Although neurons selective for global motion have also been found in mouse visual cortex, it remains unknown whether rodents can combine multiple motion signals into global, integrated percepts. To address this question, we trained two groups of rats to discriminate either gratings (G group) or plaids (i.e., superpositions of gratings with different orientations; P group) drifting horizontally along opposite directions. After the animals learned the task, we applied a visual priming paradigm, where presentation of the target stimulus was preceded by the brief presentation of either a grating or a plaid. The extent to which rat responses to the targets were biased by such prime stimuli provided a measure of the spontaneous, perceived similarity between primes and targets. We found that gratings and plaids, when used as primes, were equally effective at biasing the perception of plaid direction for the rats of the P group. Conversely, for the G group, only the gratings acted as effective prime stimuli, while the plaids failed to alter the perception of grating direction. To interpret these observations, we simulated a decision neuron reading out the representations of gratings and plaids, as conveyed by populations of either component or pattern cells (i.e., local or global motion detectors). We concluded that the findings for the P group are highly consistent with the existence of a population of pattern cells, playing a functional role similar to that demonstrated in primates. We also explored different scenarios that could explain the failure of the plaid stimuli to elicit a sizable priming magnitude for the G group. These simulations yielded testable predictions about the properties of motion representations in rodent visual cortex at the single-cell and circuitry level, thus paving the way to future neurophysiology experiments.

## Author summary

Inferring motion direction of visual objects is computationally challenging. This is because natural objects are made of multiple oriented features. Neurons in low-level visual areas, such as primary visual cortex (V1), can "see" only these local features through their

https://doi.org/10.5281/zenodo.5156156. The source data to produce all the plots presented in the main and supporting figures can be found in the Supporting Information.

**Funding:** This work was supported by a European Research Council Consolidator Grant (DZ, project n. 616803-LEARN2SEE). The funders had no role in study design, data collection and analysis, decision to publish, or preparation of the manuscript.

**Competing interests:** The authors have declared that no competing interests exist.

small receptive fields. As a result, these neurons (known as *component cells*) will report the presence of many oriented edges, each moving along a direction that is orthogonal to the edge itself. How can the brain compute the global direction of the whole object from this cacophony of disparate, local motion signals? Decades of studies in primates have shown that neurons in downstream, higher-order visual areas (known as *pattern cells*) integrate and combine motion signals encoded by component cells to represent global motion direction of the whole object. Although pattern cells have also been found in rodent visual cortex, they are so few that it is unclear whether they can support perception of global motion. In our study, we showed that rats are indeed capable of perceiving global motion direction of complex visual patterns and we verified, through computer simulations, that this ability is consistent with the representation of motion information by a population of pattern cells.

## Introduction

Extracting global motion direction of visual objects is crucial to guide behavior in many ethological contexts [1,2]. Such computation has been widely studied in the dorsal stream of primates, and, in particular, in the middle temporal area (MT) of the macaque [3]. This area is known to receive direct input from neurons in primary visual cortex (V1), which signal the "local" direction of moving elements (e.g., edges) in their small, localized, receptive fields (RF). MT units appear to combine the afferent V1 inputs in such a way to acquire selectivity for the "global" motion direction of visual objects (or patterns) made of multiple local edges [4]. The leading hypothesis emerging from the primate literature is that this is achieved by integrating the local motion signals carried by V1 afferents over different spatial positions, directions and frequencies [3–5]. Such integration is necessary, because the local output of any V1-like edge detector is intrinsically ambiguous insofar global motion is concerned. Any local output can be generated by infinite combinations of global object directions and speeds. This ambiguity is at the core of what is known, in the psychophysics literature, as the "aperture problem" [6–8]: only by combining information from multiple, local, moving-edge detectors, it is possible to infer global motion direction. Achieving a circuit level understanding of such computation remains a key question in visual neuroscience.

Over the past 10 years, the development of a wide array of tools for the dissection of neural circuits [9–13], combined with the demonstration of advanced visual behaviors in mice and rats [14,15], has fostered the use of rodents as model systems to study visual cortical processing. This led some investigators to look for the signature of motion integration in the rodent brain [16–18]. Inspired by monkey studies [19–22], they recorded the responses of mouse visual cortical neurons to drifting gratings and coherent plaids – i.e., complex visual patterns made of two overlapping gratings with the same contrast and speed, but moving along independent directions. These stimuli have been widely used to investigate motion integration in human and non-human primates since they enable to clearly distinguish local- from global-based motion responses. A plaid, in fact, has two well-defined local motion components and a global direction that is different from the local ones, while, for a grating, the local and global directions coincide [5]. By employing these stimuli, mouse visual neurons were classified as "pattern" (i.e., responsive to global motion) or "component" (i.e., responsive to local motion). The majority of neurons amenable to such classification fell into the component category, but a small fraction of pattern units was reported in V1 [17,18] and in two extrastriate areas, the lateromedial (LM) and rostrolateral (RL) areas [16]. However, the relevance of such tiny

population of pattern cells in determining mouse motion perception was left untested (but see further discussion of [17] below).

Even if a few studies tested the ability of rats and mice to discriminate the dominant motion direction of random dot kinematograms [23–27] (RDKs: a task linked to motion integration, since it requires spatial pooling of local motion cues), no study has probed the ability of rodents to perceive and report global motion direction of plaids made of extended oriented elements (e.g., gratings) moving along different directions. Notably, although both motion discrimination tasks require some form of spatial integration, processing plaids constitutes a more stringent testing ground of high-level motion integration than processing RDKs. In monkey MT, only about one-third of neurons behave like pattern cells when tested with plaids, while about 85% of MT units have been classified as pattern cells when tested with RDKs [28]. This suggests that MT neurons are better at extracting global motion direction when stimuli are broadband in orientation, as in the case of RDKs. Consistently with such easier processing of RDKs, as compared to plaids, only a tiny fraction of pattern cells has been found in mouse visual cortex, while selectivity for RDKs is way more abundant (about one-third of visually driven neurons in layers 2/3 of mouse V1) and strong enough to sustain motion discrimination accuracy of mice tested with RDKs [26].

Only in [17] mice were tested with plaids. However, rather than asking the animals to explicitly report the perceived direction of the stimuli, the authors monitored the direction of the optokinetic nystagmus (OKN) while presenting naïve mice with either plaids or gratings. The resulting bimodal distribution of OKN movements tracking both the local-components and the overall global directions of the stimulus is suggestive of mouse ability to spontaneously perceive global motion. However, this conclusion is affected by the intrinsic limitation of OKN-based experiments to probe cortical processing. In fact, OKN is a reflexive phenomenon that is known to be largely controlled by subcortical structures [29–31], while motion perception in direction discrimination tasks has been shown to rely heavily on visual cortex [26,32,33]. As a result, it remains unclear whether rodents spontaneously combine independent motion signals of extended oriented patterns (as those found in plaids) into integrated percepts of global motion direction.

Our study was designed to address this question and provide a thorough psychophysical assessment of rat ability to spontaneously perceive the global direction of drifting plaids. To this aim, we relied on a visual priming paradigm [34,35] that allowed measuring the perceptual similarity between gratings and plaids with the same global direction, without explicitly training the animals to associate these stimuli to the same response category.

## Results and discussion

### A visual priming paradigm to probe spontaneous perception of global motion direction

Rats were tested using the high-throughput behavioral rig described in [15] and previously employed in several investigations of rat object recognition by our group [34,36–40]. Briefly, the rig consists of six independent operant boxes, each equipped with a computer monitor for stimulus presentation and an array of three response ports for collection of behavioral responses. Each box bears a viewing hole on one of the walls. By protruding its head through the hole, a rat can reach with its nose/mouth the response ports and face frontally the display. The task is self-paced, in that a rat autonomously triggers stimulus presentation by licking the central response port and reports the identity of the stimulus by licking either the left or right port (details about the rig and the training procedures can be found in Materials and Methods).

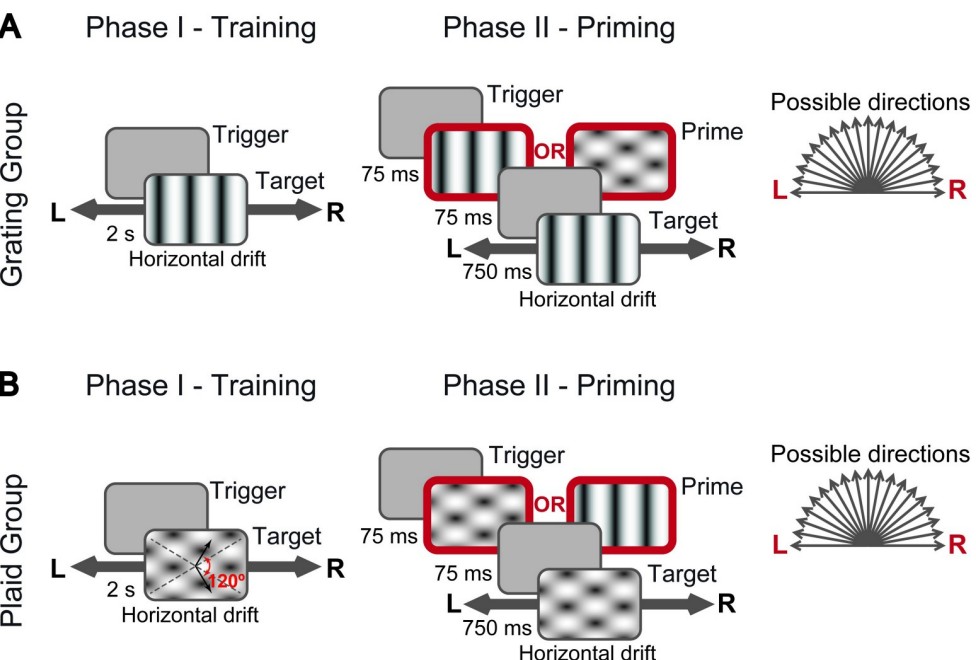

**Fig 1. Visual stimuli and discrimination tasks.** (**A**) Discriminations tasks administered to the rats of the *grating* (G) *group*. During both the training (left) and priming (right) phases, a trial started when an animal triggered the central port of a 3-way licking sensor (see Materials and Methods). During the training phase, a target grating (drifting either leftward or rightward) was presented immediately at the onset of each trial. To receive the reward, a rat had to lick the lateral port corresponding to the direction of motion of the presented target. During the priming phase, the structure of the task was similar, but at the onset of each trial, a drifting prime stimulus was presented for 75 ms, followed by a 75 ms blank screen and finally by the presentation of the target grating (lasting 750 ms). The prime stimulus could move in 19 possible directions (from 0˚ = right to 180˚ = left) and could be either a grating or a plaid. (**B**) Discriminations tasks administered to the rats of the *plaid* (P) *group*. The structure of the task was the same as in A, with the key difference that the target stimulus was a plaid instead of a grating (again drifting either leftward or rightward). In this case, the reward was delivered when a rat licked the lateral port corresponding to the global motion direction of the plaid. The plaid was made of two superimposed gratings, whose orientations are indicated by the gray, dashed lines, and whose directions are shown by the black arrows. The angle between the gratings' directions was 120˚ (red arrow).

In the first phase of the study (the *training phase*), a group of 11 male Long-Evans rats (referred to as the G group in what follows) was trained to discriminate leftward- from rightward-drifting gratings (Fig 1A, left). Another group of 10 rats (referred to as the P group in what follows) was trained to discriminate leftward- from rightward-drifting plaids with a *cross-angle* of 120˚–i.e., composite patterns made of the superposition of two gratings drifting along directions that were 120˚ apart (Fig 1B, left). Rats that maintained a criterion performance of at least 70% correct choices over a four consecutive day period were considered ready to be moved to the second phase of the study (i.e., to be tested with the priming paradigm), but were typically kept in the training phase for a few more sessions, so as to allow better consolidation of the task.

All rats starting the training phase managed to reach such criterion, although with a variable learning time. Most animals needed between 10 and 25 sessions before fully acquiring the task, while a few rats required up to 60 sessions (see Fig 2A, where the dots on the learning curves indicate when each animal reached the criterion performance). The learning rate was not significantly different between the two groups, as it can be appreciated by looking at the group average learning curves (Fig 2B) and the average numbers of sessions to criterion (Fig 2C; p > 0.05, unpaired, two-tailed t-test). The discrimination performances reached by the animals in the two groups were also very similar–although the mean performance was slightly

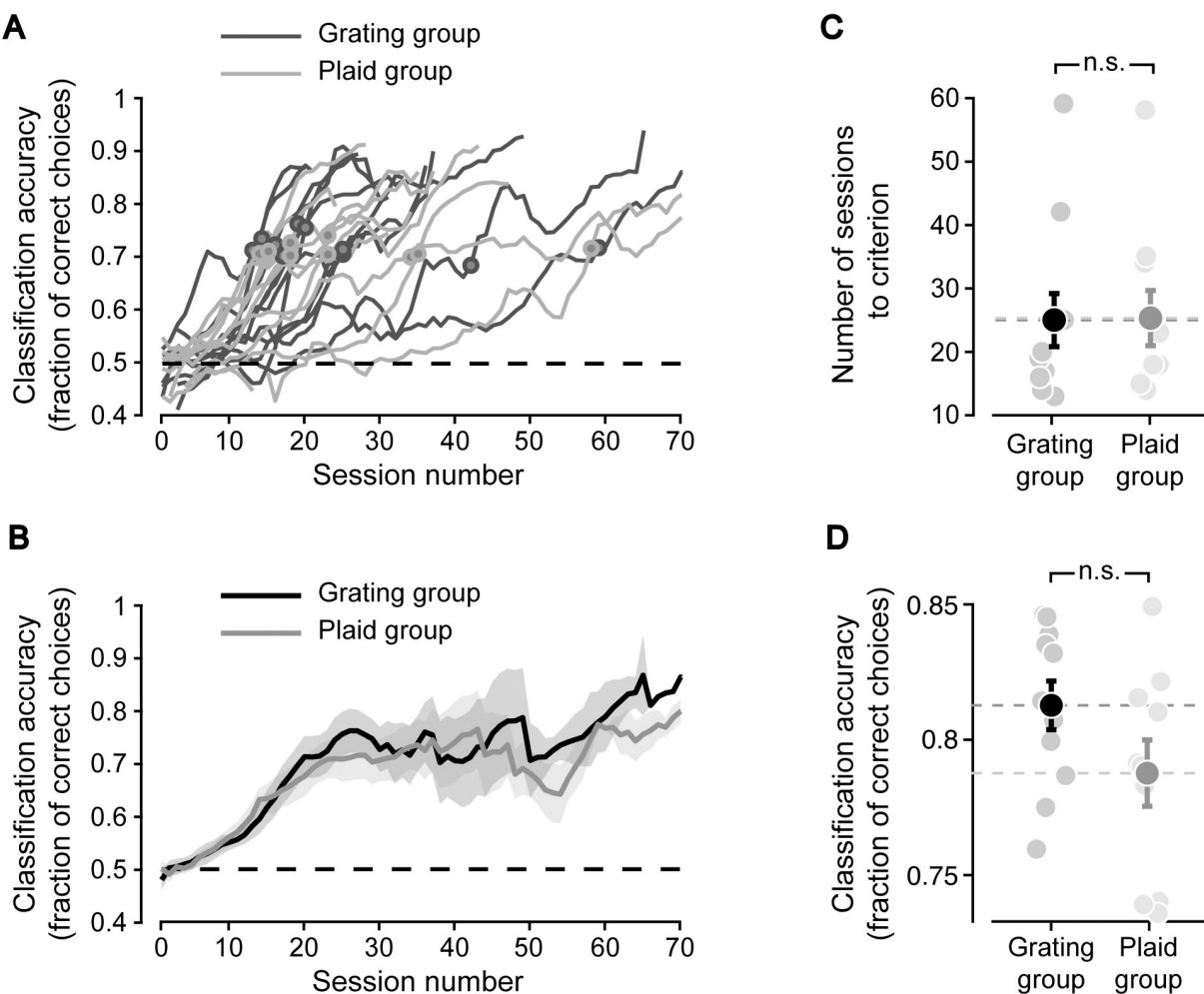

**Fig 2. Learning the motion discrimination task.** (**A**) The discrimination accuracies achieved by all the rats in the G and P groups (dark and light gray curves, respectively) are plotted as a function of the training sessions. Each value on a curve reports the mean of the performances achieved by the animal in the corresponding session and in the previous and following sessions (i.e., the curves are the result of applying a moving-average filter with size 3 to the session-by-session accuracies). The dots indicate the sessions in which each rat achieved the criterion performance of 70% correct discrimination over four consecutive days (each dot marks the first of such sessions; dark and light gray dots refer, respectively, to the animals of the G and P groups). (**B**) Group average discrimination accuracies (solid lines) as a function of the training sessions. The black and gray curves refer, respectively, to the rats of the G and P groups. The shaded regions are SE. (**C**) The numbers of training sessions required by the animals in the two groups to reach the criterion performance (pale dots; same data as those already shown in A) are reported along with their averages (dark dots; error bars are SE). Averages were not significantly different between the two groups (p > 0.05, unpaired, two-tailed t-test). (**D**) The asymptotic performances (as computed by averaging the discrimination accuracies in the session when criterion was reached and any ensuing session) of the animals in a group (pales dots) are reported along with their average (dark dots; error bars are SE). Averages were not significantly different between the two groups (p > 0.05, unpaired, two-tailed t-test).

higher for the G group than for the P group (Fig 2D; 81% vs. 79% correct), the difference was not significant (p > 0.05, unpaired, two-tailed t-test). This indicates that rats are equally capable of discriminating motion direction of drifting gratings and plaids.

In the priming paradigm, the "target" stimulus remained the same as in the training phase (i.e., G and P rats still had to report the direction of drifting gratings and plaids, respectively), but now the presentation of the target was preceded by the brief presentation (75 ms) of either a grating or a plaid drifting along one of 19 possible directions (from 0˚ = rightward to 180˚ = leftward, in steps of 10˚; Fig 1A and 1B, right). The identity (i.e., either grating or plaid) and the motion direction of such "prime" stimulus was randomly selected in each trial. In what

follows, we refer to trials in which the identities of prime and target coincide as *identity-priming* condition (e.g., a grating prime stimulus followed by a grating target stimulus), whereas we refer to the opposite case as *cross-priming* condition (e.g., a plaid prime stimulus followed by a grating target stimulus). Critically, the rats kept receiving feedback (i.e., either reward or a time-out period in case of correct and incorrect choices, respectively) only about the correctness of their responses to the target stimuli. The identity and direction of the prime stimuli were never paired to either the leftward or rightward response categories. As such, the extent to which the prime stimuli were able to affect the choices of the rats was purely due to the spontaneous, perceived similarity between primes and targets. Assessing such similarity was the ultimate goal of our experiments, since it allowed understanding whether, in the cross-priming condition, the perceived motion direction of the plaids was the global one. Based on previous motion adaptation studies in humans [41] and on former priming/masking experiments carried out in rats [34,42–44], we choose the duration of the prime and of the inter-stimulus interval (ISI) between prime and target (75 ms each) in the attempt of inducing a strong priming effect–i.e., in the attempt of biasing rats' responses to the target towards the motion direction of the prime.

In what follows, we will report the accuracy of rats at discriminating the motion direction of a target stimulus as a function of its angular distance from the direction of the prime stimulus that preceded it. As such, the resulting *priming curve* will range between two extremes: 1) the case in which the direction of the prime and target stimuli was the same (e.g., both drifting rightward), referred to as *coherent prime* condition; and 2) the case in which the prime and target stimuli had opposite directions (e.g., the prime drifting rightward and the target drifting leftward), referred to as *incoherent prime* condition. The case in which the prime stimulus drifted vertically (midpoint of the priming curve) will be referred to as *neutral prime* condition. This terminology will be used for both the identity- and cross-priming conditions and for both the P and G groups.

The first step in our analysis was to verify whether the prime stimuli were indeed capable of biasing rat choices depending on the level of coherence between their direction and that of the target stimuli, in case of the identity-priming condition. This was the case for 10 out of 11 rats of the G group and 9 out of 10 rats of the P group, which were therefore included in the subsequent analysis (see Materials and Methods). For the G group, rat group average accuracy at discriminating the motion direction of the target gratings was strongly modulated, as function of the direction of the prime grating stimulus (Fig 3A, black curve). In the coherent prime condition, rat classification of target direction was facilitated, as compared to the reference neutral prime condition (dashed line). Conversely, in the incoherent prime condition, rat classification accuracy was substantially lowered. Overall, the identity-priming curve was approximately sigmoidal. Even for the P group, in the identity-priming condition, a strong priming was observed (Fig 3B, black curve). Also in this case, the magnitude and sign of the priming effect depended on the similarity between the directions of the prime and target plaid stimuli, resulting in an approximately monotonic drop of rat classification accuracy from the coherent priming condition towards the incoherent one. The overall priming magnitude observed in the identity-priming condition (black bars in the insets of Fig 3A and 3B) was virtually identical for the two groups, resulting in no difference between priming magnitudes (S1 Fig, left dot). The same applied to the difference between the performances observed for the two groups in the case of the neutral prime stimuli (S1 Fig, right dot). Along with the results of the training phase (Fig 2), this confirmed that the rats in the two groups were similarly sensitive to the direction of the class of stimuli they were trained with (i.e., the gratings for the G group, and the plaids for the P group).

The strong modulation of rat classification accuracy in the identity-priming condition has two important implications. First, it demonstrates the effectiveness of the priming paradigm in capturing the perceived similarity between the learned (horizontal) motion directions of the

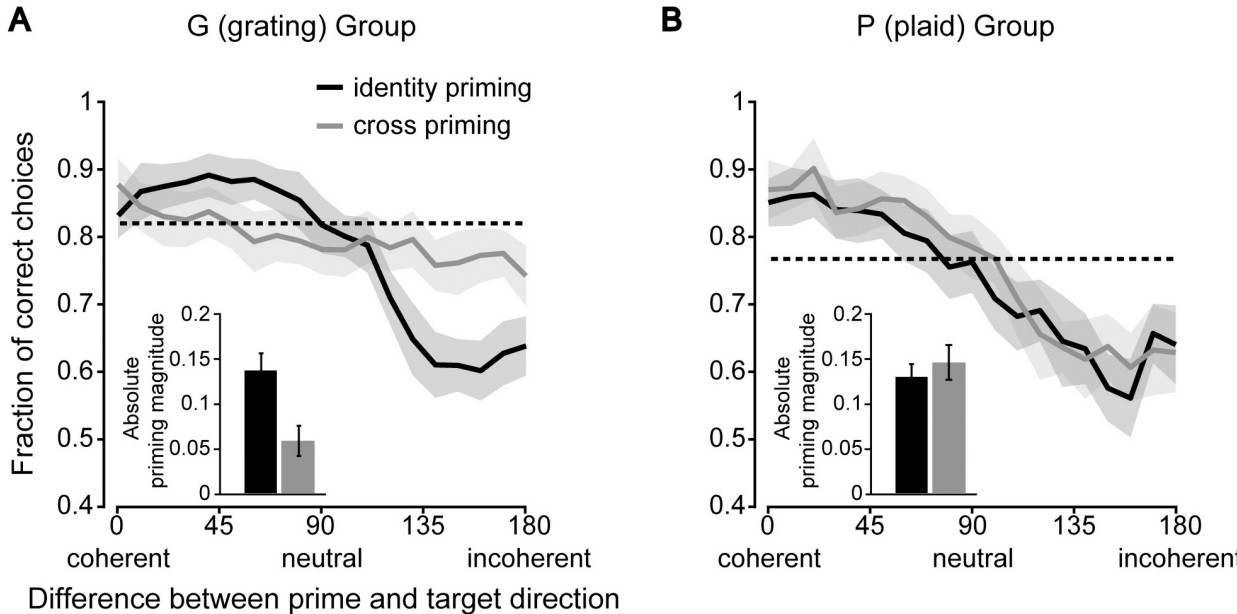

**Fig 3. Identity- and cross-priming curves.** (**A**) The identity-priming curve (black) reports the mean accuracy of the rats of the G group at discriminating the leftward from the rightward drifting gratings, as a function of the difference between the direction of these target stimuli and the gratings that were used as prime stimuli. The cross-priming curve (gray) reports rat mean accuracy when the plaids were used instead as prime stimuli. The shaded areas are 95% confidence intervals obtained by bootstrap (see Materials and Methods). The dashed line shows the accuracy achieved with the neutral identity-prime stimulus (i.e., the grating drifting upward). The inset quantifies the absolute magnitude of priming obtained in the two cases, as the mean absolute difference between the four leftmost (and rightmost) points in a curve and the point corresponding to the neutral condition. The error bars show 95% confidence intervals obtained by bootstrap (see Materials and Methods). (**B**) Same as in A, but for the rats of the P group. That is, in this case, the curves report rat mean accuracy at discriminating the leftward from the rightward drifting plaids. The identity-priming curve (black) refers to trials where the plaids were used as prime stimuli, while the cross-priming curve (gray) refers to trials where the gratings were used as primes. S1 Fig shows that neither the priming magnitudes (i.e., the black bars in the insets of A and B), nor the accuracies corresponding to the neutral primes were significantly different for the rats of the two groups.

target stimuli and the untrained (previously unseen) directions of the prime stimuli, in the case in which targets and primes are either both gratings or both plaids (i.e. *identity-priming* condition). Second, and more importantly, it serves as a reference against which to compare the modulation of the accuracy observed in the cross-priming condition. In fact, the degree of similarity between the priming curves measured in the identity- and cross-priming conditions (i.e., between the black and gray curves in Fig 3) allows answering two key questions about the neuronal representations of the grating and plaid stimuli.

## Evidence of a shared representation of gratings' and plaids' motion direction

The first question is whether these representations are "shared" (i.e., the same neural population encodes both gratings and plaids) or "non-shared" (i.e., independent, non-overlapping populations represent the two types of stimuli). In the former case, we would expect a substantial modulation of rat classification accuracy also in the cross-priming condition, possibly as large as in the identity-priming condition. In the latter case, no cross-priming effect would be observable, given the independence of the two populations underlying the perception of gratings and plaids. In case of shared representations, then a second crucial question could be answered by comparing the shapes of the priming curves obtained in the identity- and cross-priming conditions. If rats spontaneously represented the stimuli according to their global direction, then the priming curves produced by gratings and plaids would be very similar,

given that, by construction, the global direction of the prime stimuli was the same for matching grating and plaid primes. If the opposite were true, then the priming curves produced by gratings and plaids would be different.

In the G group (Fig 2A), the cross-priming curve (gray) was almost flat and significantly different from the identity-priming curve (black), as shown by the non-overlapping 95% confidence intervals (shaded areas). The priming magnitude produced by the plaid stimuli was found to be significantly smaller than that produced by the gratings (Fig 2A, inset: gray vs. black bar; non-overlapping 95% confidence intervals). This inability of the plaids (when used as prime stimuli) to affect the discrimination (hence the perception) of the target gratings suggests that, for the rats of the G group, the neuronal population representing the direction of the gratings was poorly activated by the presentation of the plaids.

By contrast, in the P group, the cross-priming and identity-priming curves largely overlapped (Fig 2B: gray vs. black curve). As a result, the priming magnitude observed in the two conditions was equally large (Fig 2B, inset: gray vs. black bar; overlapping 95% confidence intervals). In other words, the gratings and plaids (when used as prime stimuli) were equally effective at biasing rat discrimination of the target plaids. This indicates that the same neuronal population was active during the presentation of both kinds of stimuli, as expected in case of a shared representation of gratings and plaids. More importantly, the priming produced by gratings and plaids was not only equivalent in terms of overall magnitude, but it was also very similar for matching (global) directions of the two stimuli (i.e., overlapping curves in Fig 2B). This strongly suggests that the representation of plaids in rats trained to discriminate such stimuli (P group) is not only shared with the representation of gratings but is structured in such a way to encode plaids' global motion direction.

## Rat perception of drifting plaids is more consistent with a pattern- than a component-based representation of motion signals

To test more formally the extent to which the priming curves observed for the P group are consistent with a representation of global motion direction of the plaid stimuli, we performed a decoding analysis using linear classifiers. We simulated two alternative scenarios: a representation made of pattern cells (i.e., neurons tuned to the global direction of the plaids) and a representation made of component cells (i.e., neurons tuned to the local direction of the constituent gratings of the plaids). While it may seem obvious that only a pattern-based representation can underly the results of Fig 3B, the use of a low-level, component-based representation cannot be ruled out without further analysis for at least three reasons. First, rats are capable of solving apparently complex visual tasks (e.g., object recognition) using very simple, low-level strategies (e.g., measuring global luminance differences among the objects) [45,46], unless the stimuli and tasks are carefully designed to specifically engage higher-order, shape-based processing [36–38]. The same applies to visual object representations in V1, which can support recognition of visual objects despite image-level transformations (e.g., translation, scaling and rotation) to a surprising extent [47–49]. This is because they can rely on mean luminance differences among the objects–only if such differences are leveled out, higher-order extrastriate areas become necessary to support invariant object recognition [47]. Second, as already mentioned in the Introduction, evidence about the existence of pattern cells in rodents is so sparse [16–18] that it is questionable whether those few pattern cells reported in previous studies have any role in mediating rodent motion perception. Finally, previous monkey studies have pointed out that component cells may actually display a pattern-like behavior if direction tuning is broad enough [50]. Hence the need of exploring which scenario can yield the virtually undistinguishable identity- and cross-priming curves of Fig 3B.

Both the component- and the pattern-based representations were simulated as a population of 24 units, whose preferred directions homogeneously spanned the [0° 360°] range (i.e., one unit tuned to every direction in steps of 15°; Fig 4A and 4B). The tuning curves of the simulated units were defined by Von Mises functions [51,52], which are the circular analog of Gaussian functions and whose width is controlled by the parameter $k$, which is equivalent to the inverse of the variance of a Gaussian distribution [53] (see Materials and Methods for details). In our simulations, $k$ was set to 7, which roughly corresponds to a tuning curve with a full width at half maximum (FWHM) of about 50° (see grayscale matrix in Fig 4A and 4B). By construction, for a component cell, the response to the drifting plaids as a function of their direction featured two peaks, separated by 120° (Fig 4A, bottom). These peaks correspond to the global directions of the plaid that bring one of the constituent gratings to be aligned to the preferred (local) direction of the unit (i.e., the one shown in Fig 4A, top). Conversely, the tuning curves of the simulated pattern cells featured a single peak, corresponding the global direction of the stimulus (no matter whether grating or plaid) they were tuned to (Fig 4B) (see Materials and Methods for further details).

We used these populations to simulate how a rat trained to discriminate leftward from rightward drifting plaids (i.e., same task of the animals in the P group; see Fig 1B) would spontaneously categorize the novel, previously unseen gratings and plaids used as prime stimuli, under the assumption of either a purely pattern-based or a purely component-based representation. Rat perceptual decisions were simulated by training a logistic classifier to discriminate the leftward from the rightward drifting plaids, based on their representation in the simulated neural population space. As a result of the training, each classifier learned to rely on a specific subset of the units of the representation, as shown by the weights' distribution of Fig 4C and 4E. We then tested how each classifier categorized the sets of drifting plaids and gratings used in the identity- and cross-priming experiments. This yielded generalization curves (Fig 4D and 4F) that, in the [0° 180°] range, are equivalent to the identity- and cross-priming curves of Fig 3B (to ease the comparison with the latter, the right side of the curves in Fig 4D and 4F is grayed out).

The shape of the generalization curves can be understood by considering the corresponding weights' distribution resulting from training with the 0°- and 180°-drifting plaids. In the case of the component representation, these training stimuli activated mostly the units tuned for directions at ±60° with respect to the direction of the stimuli themselves. As a result, the classifier learned to assign large, positive weights to the units tuned at ~60° and ~300° (i.e., those providing the strongest evidence of a rightward, 0°-drifting plaid), and large, negative weights to the units tuned at ~120° and ~240° (i.e., those providing the strongest evidence of a leftward, 180°-drifting plaid)–hence, the twin positive and negative peaks in the weights' distribution (Fig 4C).

When the classifier was presented with test plaids spanning the whole [0° 360°] range of directions, the proportion of rightward choices (solid curve in Fig 4D) was obviously maximal (i.e., 100%) at 0° (since this was the rightward training direction), but then dropped quickly, becoming minimal (i.e., 0%) for a plaid drifting at 60°. In fact, for this stimulus, one of the constituent gratings activated the units tuned at ~0°, where the weights were close to zero (see Fig 4C), while the other grating activated the units tuned at ~120°, where the weights were large but negative, since those units signaled the presence of the leftward, 180°-drifting plaid. When the direction of the test plaid further increased, the proportion of rightward choices grew again, reaching a new positive peak at 120°. This was because one of the constituent gratings of the plaid activated the units tuned at 60° (with large, positive weights), while the other grating activated those tuned at 180° (where the weights were close to zero). The generalization curve reached again another minimum when the test plaid drifted at 180° (the leftward training

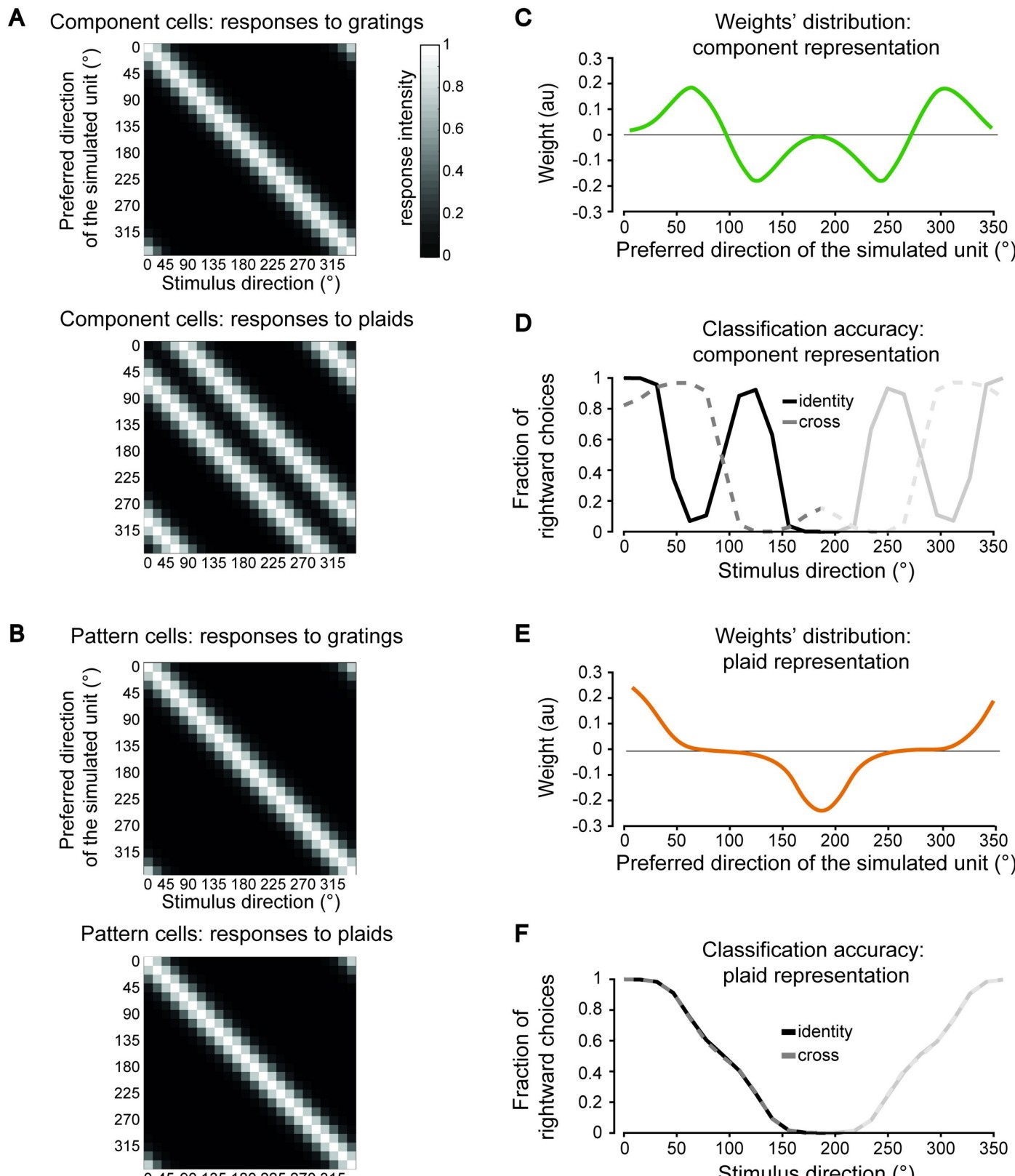

**Fig 4. Simulating the discrimination accuracy afforded by component- and a pattern-based representations for rats trained to discriminate drifting plaids.**
(**A**) Tuning curves (rows) for a simulated population of 24 components cells, whose preferred directions homogeneously spanned the [0˚ 360˚] range, starting from

0˚ (first row), up to 345˚ (last row), in steps of 15˚. The responses of the units are shown as a function of 24 possible directions (columns) of the grating (top) and plaid (bottom) stimuli. The gray scale bar indicates the intensity of the response. (**B**) Same as in A but for a population of 24 simulated pattern cells. (**C**) Weights assigned to each of the 24 component cells in A (the x axis reports the preferred direction of each unit) by a logistic classifier that was trained to discriminate a 0˚-(rightward) from a 180˚-drifting (leftward) plaid. (**D**) Fraction of times that the logistic classifier fed with the component-based representation (whose weights are shown in C) classified the test stimuli as drifting rightward (i.e., at 0˚). The test stimuli were either plaids (solid black curve) or gratings (dashed gray curve) spanning the whole [0˚ 360˚] range of directions. (**E**) Weights assigned to each of the 24 pattern cells in B by a logistic classifier that was trained to discriminate a 0˚-(rightward) from a 180˚-drifting (leftward) plaid. (**F**) Same as in D, but for the logistic classifier fed with the pattern-based representation (whose weights are shown in E). Note that, by construction, the curves in C-F are symmetrical around the middle point on the abscissa (180˚). To emphasize that the leftward halves of the curves in D and F correspond (and should be compared) to the curves with matching colors in Fig 3B, the rightward sides of D and F were grayed out. S2 Fig reports an extended version of these simulations, where a range of widths of the tuning curves (from very broad to very narrow) was tested. S3 Fig reports a version of these simulations where gratings instead of plaids were used as training stimuli.

direction), with the constituent gratings that activated the units tuned at 120˚ and 240˚, with large, negative weights. As shown in Fig 4D, a mirror trend was produced for plaids drifting along directions larger than 180˚ (grayed out portion of the curves).

As for the generalization curve obtained when feeding to the classifier isolated drifting gratings (dashed, gray curve in Fig 4D, which is equivalent to the cross-priming curve of Fig 3B), it largely followed the shape of the weights' distribution. This can be easily understood, given that a grating provided as much evidence of rightward (leftward) motion as close its direction was to the positive (negative) peaks of the weights' distribution. As a result, the generalization curves produced by plaids and gratings (solid vs. dashed curve in Fig 4D) followed completely different trends, with no overlap. In addition, none of them matched the monotonic decrease observed for the corresponding identity- and cross-priming curves of Fig 3B. Together, this strongly suggests that a representation made of component cells is incompatible with the priming curves measured for the rats of the P group.

A very different result was found for the simulated representation of pattern cells. In this case, the weights' distribution had a large, positive peak at 0˚, corresponding to the units coding for the rightward motion, as well as a large, negative peak at 180˚, corresponding to the units coding for the leftward motion (Fig 4E). Given that, by construction, the pattern cells responded in the same way to isolated gratings and plaids having the same (global) direction, the generalization curves corresponding to the identity- and cross-priming conditions fully overlapped (Fig 4F, black, solid line vs. gray, dashed line). More importantly, they both followed a monotonic, decreasing trend in the [0˚ 180˚] range, which largely trailed the shape of the weights' distribution. These trends were highly consistent with those observed for the priming curves of Fig 3B. This shows that, differently from the case of the component representation, a representation made of pattern cells would support the identity- and cross-priming curves obtained for the rats of the P group.

To verify the extent to which this conclusion depended on the width of the tuning curves of the simulated units, we allowed the parameter $k$ (that defines the width of the Von Mises functions) to range from 0.5 to 11. Examples of the resulting tuning curves obtained for the simulated component cells are shown in S2A Fig. At very low $k$, as a result of the increased broadness of the tuning, the two peaks in the response to the plaid stimuli (solid curves) tended to merge. This translated in a broadening of the weights' distribution of the logistic classifier (S2B Fig) and in a concomitant increase of the similarity between the generalization curves obtained for gratings and plaids (S2C Fig; dashed vs. solid curves). To quantify this similarity, we computed the absolute differences between the proportions of rightward choices in the two generalization curves across all tested directions and we averaged them. The resulting average difference was large (as already shown in Fig 4D) for $k > 5$ but became negligible for $k < 3$ (S2D Fig). This corresponds to peaks in the tuning curves with FWHM ~ 80˚ or wider (see S2A Fig). Therefore, a population of component cells with such a broad tuning could in

principle account for the overlapping identity- and cross-priming curves obtained for the P group in our experiments (Fig 3B).

To understand whether this is a plausible scenario, one would need to measure the width of tuning of component cells in rodent visual cortex. Unfortunately, the few studies investigating component and pattern cells in mouse visual cortex did not perform a statistical characterization of the sharpness of their orientation tuning. This prevented our simulated units from quantitatively matching a physiological ground-truth in terms of orientation tuning. Nevertheless, the tuning curves of the example component cells reported in rodent studies investigating this topic feature such sharp and narrow peaks over the orientation axis to decisively fall in the regime corresponding to $k >> 5$ in our simulations (i.e., with FWHM $<< 60°$) [16,17]. A similar conclusion is supported by the general tendency of orientation tuning to be quite sharp in mouse and rat visual cortex [49,52,54–59]. As such, the most likely hypothesis is that the stimulus representation underlying the priming curves of the P group (Fig 3B) consisted of pattern, rather than component cells. Further neurophysiological studies will be necessary to verify this hypothesis–e.g., by directly testing, as done here in our simulations, the extent to which populations of recorded component and pattern cells yield direction classification curves that are consistent with the priming curves measured in our experiments.

For completeness, we also simulated a scenario where the training stimuli were gratings instead of plaids. That it, we checked whether the priming curves obtained for the G group could possibly be explained by a purely component- or a purely pattern-based representation. As shown in S3A Fig, in the case of a simulated representation of component cells, the classifier learned to discriminate 0°- from 180°-drifting gratings by assigning large, positive weights to the units tuned at ~0° and large, negative weights to the units tuned at ~180°. As a result, when tested with grating directions spanning the whole circle, the classifier accuracy trailed the shape of the weights' distribution (S3B Fig, black, solid line), following an approximately sigmoidal trend in the [0° 180°] range. When tested with drifting plaids, the classifier accuracy peaked instead at 60° and reached a minimum at 120°, consistently with the 120° cross-angle of the plaids (S3B Fig, gray, dashed line). In the case of a simulated pattern representation, the behavior of the classifier was identical to the one already observed for the simulated P group (Fig 4E and 4F). The weights were maximal at 0° and minimal at 180° (S3C Fig) and the accuracy curves were the same no matter whether the classifier was tested with gratings or plaids (S3D Fig, black vs. gray line), following a sigmoidal trend in the [0° 180°] range.

In conclusion, in the case of the G group, while both the component- and the pattern-based representations could account for the shape of the identity-priming curve (Fig 3A, black line), neither of them could properly explain the shape of the cross-priming curve (Fig 3A, gray line) or, better, the lack of cross-priming. As explored in the next sections, other tuning or reading out mechanisms need to be invoked to account for this phenomenon.

## Summary of our findings

The results presented in Fig 3B suggest that rats are capable of spontaneously combining local motion cues into integrated percepts of global motion direction of a complex visual pattern. Our simulations show that this ability is consistent with a representation of global motion signals, as the one provided by pattern cells (Figs 4 and S2). As such, our findings strongly suggest that rats are capable of processing motion stimuli by relying on pattern-like representations. This, in turn, establishes a candidate perceptual correlate of the neuronal selectivity for global motion previously reported across rodent visual cortical areas [16–18], suggesting that, despite being a tiny fraction of the overall visual cortical population, rodent pattern cells may play an important (possibly preferential) role in the processing of visual motion. However, such

motion integration ability only emerged in rats trained to discriminate the plaid stimuli (P group). The animals trained in the grating discrimination task (G group) were instead virtually insensitive to the plaids (Fig 3A) and this finding could neither be explained by simulating a component- nor a pattern-based representation (S3 Fig).

Critically, the different cross-priming magnitude observed for the rats of the two groups cannot be explained by an overall lower discriminability of the plaid stimuli, i.e., by a lower sensitivity of rats to plaids as compared to gratings. As already explained, our experiments show that both kinds of stimuli are equally well processed by rats, with the G and P groups reaching statistically undistinguishable performances in the training phase of our study, both in terms of learning rate and discrimination accuracy (Fig 2). In addition, both kinds of stimuli (gratings and plaids) were equally effective when used as primes in the identity priming tests (compare black curves and bars in Fig 3A and 3B), yielding statically undistinguishable priming magnitudes (S1 Fig). In other words, the failure of the plaid stimuli to affect the discrimination of the gratings for the G group cannot possibly be accounted for by a general inability of rats to perceive, process or discriminate drifting plaids. Our fully counterbalanced design, with the high plaid discrimination performance (Fig 2) and the strong priming effect induced by the plaids in the P group (Figs 3 and S1), implies that the insensitivity of rats in the G group to the plaids depended on their training history. This suggests a training-dependent recruitment of different populations of motion detectors in the two groups, with the neuronal population relied upon by the rats of the G group to encode the gratings failing to effectively represent the plaids.

In the following sections we provide two tentative explanations of such observations based on different sets of assumptions. Simulations of component- and pattern-based representations and linear read-out mechanisms (similar to those shown in Fig 4) allowed testing the ability of such assumptions to possibly explain our findings.

## Simulating a scenario where training with either gratings or plaids leads to a decoding pool enriched of either component or pattern cells

The first explanation rests on the hypothesis that the specific discrimination being reinforced during the training biased the recruitment of the visual cortical pool that was read out by downstream decision neurons towards either a more component-enriched or a more pattern-enriched cell population (referred to as the "decoding pool" in what follows). Such a task-dependent selection of the decoding pool may stem from differences between component and pattern cells in terms of two key properties: i) their relative proportion within visual cortex; and ii) the intensity of their responses to gratings and plaids.

With regard to the first property, although the number of studies investigating motion integration in rodent visual cortical areas is still limited [16–18], a finding that is consistent among them is that component cells by far outnumber pattern cells, with the former being ~25% of direction-tuned units and the latter being about ~5% (with the rest being unclassified). With respect to the second property, a well-known phenomenon likely affecting the strength of grating and plaid responses is cross-orientation suppression [60–63], which has been recently documented also in rodents [42,64,65]. It consists in a reduction of the response to the preferred oriented stimulus when this is presented together with other stimuli with different orientations. This means that cross-orientation suppression could, in principle, lead to diminished responses to plaids as compared to gratings. However, most studies documenting this phenomenon mainly targeted V1, without distinguishing between pattern and component cells. In addition, when focusing on studies of motion integration in rodents [16–18], it was difficult to draw any insight about the incidence of cross-orientation suppression in pattern and

component cells, because of methodological limitations (e.g., tuning curves normalization and calcium imaging lacking spike-count-level information). This is why we turned to the primate literature, finding hints of cross-orientation suppression being stronger in component cells than in pattern cells [20,21].

Our hypothesis is that the combined effect of component cells being more numerous than pattern cells, but also more sensitive to cross-orientation suppression, led the decoding pool to become dominated by component cells in the case of training with gratings (G group) and by pattern cells in the case of training with plaids (P group). In turn, this would qualitatively explain the difference in terms of cross-priming found for the rats of the G and P groups (Fig 3), as graphically illustrated in Fig 5A and 5B. Briefly, in the case of training with gratings (Fig 5A, top), these stimuli would strongly activate both pattern (orange) and component (green) cells (filled and unfilled neurons indicate, respectively, responsive and unresponsive units). As such, the selection of the neuronal decoding pool by a downstream decision neuron would not be biased towards either class of neurons (because of no differences in their responsiveness). However, since the component cells substantially outnumber the pattern cells, the decoding pool would contain more of the former (black-boundary neurons) than of the latter (red-boundary neurons). This follows from the assumption that the selection of the decoding pool is the result of a random sampling from the subpopulation of all responsive units. This, in turn, is consistent with the idea that such recruiting process is mediated by some form of plasticity similar to the reward-gated Hebbian learning reviewed in [66]. In the priming experiment, all the neurons in the decoding pool would be active during presentation of the grating primes (Fig 5A, middle), leading to the strong identity priming observed in Fig 3A (solid curve). By contrast, most component cells in the pool would be strongly depressed by cross-orientation suppression during presentation of the plaid primes (Fig 5A, bottom; empty black-boundary neurons), and the few pattern cells in the pool (orange-filled, red-boundary neurons), although active, would not be enough to affect the perception of the target stimulus. Hence, the lack of cross-priming in the group of rats trained with gratings (Fig 3A, gray curve).

In the case of training with plaids (Fig 5B, top), these stimuli would strongly activate the population of pattern cells (orange-filled neurons) but only weakly excite the component cells (green-filled neurons), because of cross-orientation suppression. This would lead to a strong bias towards the recruitment of pattern cells in the decoding pool (red-boundary neurons) by a downstream decision neuron. When tested with prime stimuli, such pattern-enriched pool would respond with a similar intensity to both kinds of primes (grating and plaids; Fig 5B, middle and bottom panels respectively), leading to similar priming curves in both identity- and cross-priming conditions, as shown in Fig 3B (solid vs. gray curve). This would account for the shared representation of global motion direction observed for rats trained with plaids.

To quantitatively test whether the mechanistic account hypothesized in the previous paragraphs could explain our findings, we built a computational model where a decision neuron had to select his afferent units from a mixed population of component and pattern cells, in order to discriminate the direction of either drifting gratings or plaids. The relative proportion of simulated component (80%) and pattern (20%) cells roughly matched that found, on average, across rodent visual areas V1, LM and RL [16–18]. Each unit was simulated as a Poisson spiking neuron, whose average firing rate as a function of direction was defined by a von Mises function [51,52] having a peak that was randomly sampled across the [0˚ 350˚] direction axis (see Materials and Methods for details).

In addition, to simulate cross-orientation suppression in component cells, their peak responses to plaids were set to half their peak responses to gratings. On the other hand, pattern cells were assumed to be unaffected by cross-orientation suppression. As for the simulation of

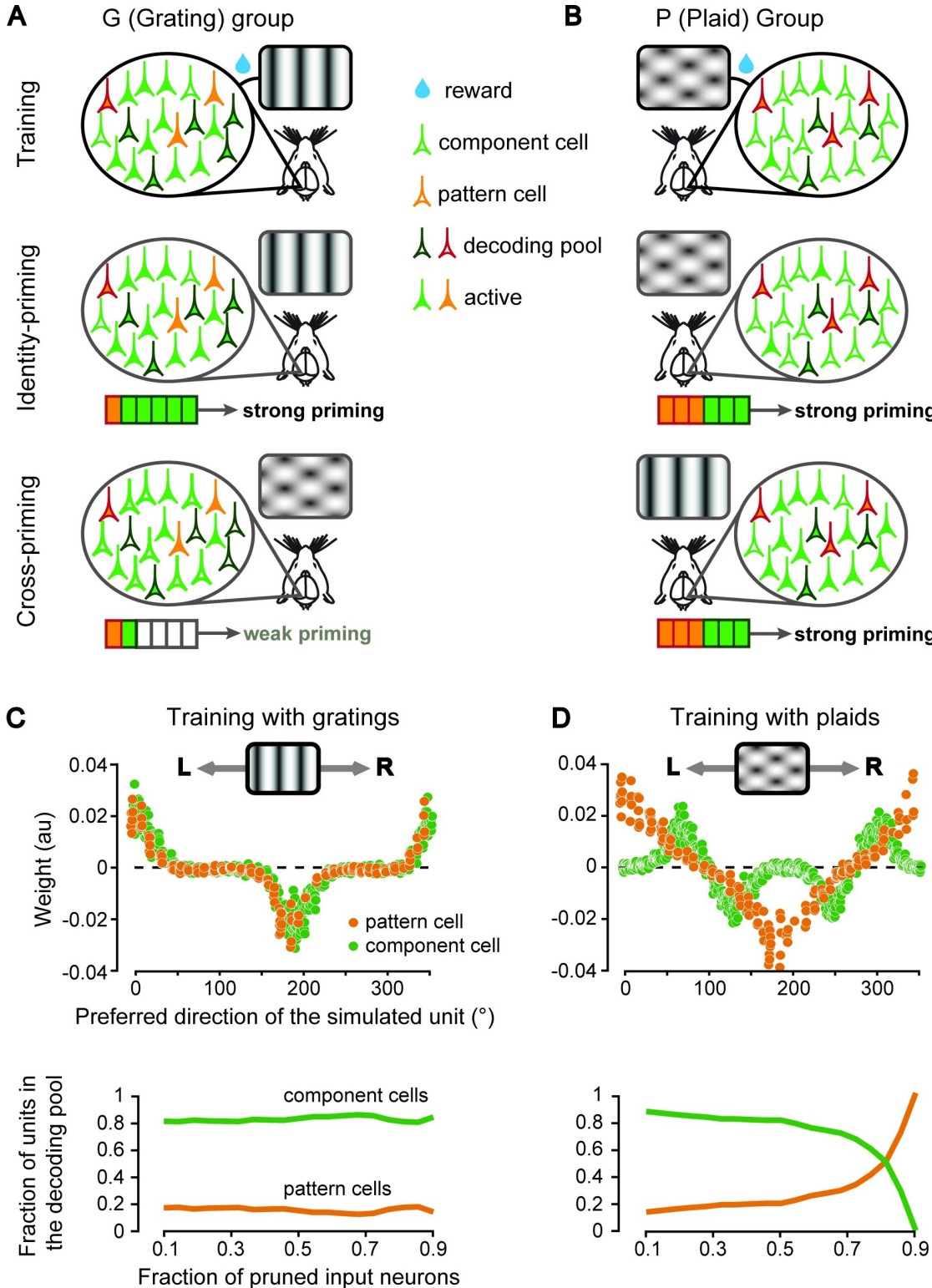

**Fig 5. Simulating a scenario where training with either gratings or plaids leads to a decoding pool enriched of either component or pattern cells.** (**A**) Cartoon illustrating how training with gratings would lead to the selection of a decoding pool enriched of component cells, under the assumption that: 1) component cells are more numerous than pattern cells; and 2) cross-orientation suppression is stronger for component than pattern cells. The sketched pyramidal neurons represent either component (green) or pattern (orange) cells. Filled and unfilled neurons indicate, respectively, responsive and unresponsive units. Black- and

red-boundary neurons indicate the cells that the decision neuron learned to rely upon to discriminate the rightward- from the leftward-drifting gratings (i.e., the cells in the decoding pool). The grating stimuli activate equally strongly the component and pattern cells, but, being the former more numerous, the decoding pool is mainly composed of component cells (top). When tested with gratings, the cells in the decoding pool are strongly activated (middle), but, when they are tested with plaids, they are only weakly activated (bottom), because of cross-orientation suppression. (**B**) Cartoon illustrating how training with plaids would lead to the selection of a decoding pool enriched of pattern cells. Because of cross-orientation suppression, component cells are weakly activated by plaids, thus the decoding pool is enriched of pattern cells (top). As a result, the decoding pool is strongly activated when presented with both plaids (middle) or gratings (bottom). (**C**) Top: weights assigned to a mixed population of component (green) and pattern (orange) cells, as a function of their preferred direction, by a logistic classifier that was trained to discriminate a 0˚- (rightward) from a 180˚-drifting (leftward) grating. Bottom: The proportion of component (green) and pattern cells (orange), initially set to, respectively, 80% and 20% of the total, remained unchanged when weights with increasingly larger magnitude were progressively set to zero (i.e., pruned from the decoding pool). (**D**) Top: same as in C (top) but for a logistic classifier trained to discriminate a 0˚- (rightward) from a 180˚-drifting (leftward) plaid. Because of cross-orientation suppression, the classifier learned to assign weights with lower magnitude to the component cells. Bottom: as a result, when the input connections to the classifier were made increasingly sparser (by pruning weights with progressively larger magnitude), the proportion of pattern cells in the decoding pool (orange) reached and overtook the proportion of component cells (green). As shown in S4 Fig, this reversal in the proportion of pattern and component cells did not take place if the latter were not affected by cross-orientation suppression.

Fig 4, the decision neuron consisted in a logistic classifier that was trained to discriminate between 0˚- and 180˚-drifting stimuli (i.e., leftward vs. rightward motion). This classifier simply performed a weighted sum over the inputs provided by all the simulated component and pattern cells, followed by the application of a sigmoidal nonlinearity. As a result of training, the weights became adjusted in such a way to maximize the classification accuracy (eventually attaining 100% in such a simple motion discrimination task).

When trained in the grating discrimination task (to simulate the training regime of the G group), the classifier learned to rely more heavily on those motion detectors (no matter whether component or pattern cells) whose preferred direction was close to that of the training stimuli. This is shown in Fig 5C (top), where the weights of the simulated units as a function of their preferred direction are reported. For both the component (green dots) and pattern (orange dots) cell populations, the most informative neurons were those with preferred direction close to 0˚ (rightward-drifting gratings) and 180˚ (leftward-drifting gratings), while the units tuned around the vertical drift direction (i.e., 90˚ and 270˚) were assigned close-to-zero weights. Given these weight distributions, we simulated different degrees of sparsity constraints on the connectivity of the decision neuron. This was done by pruning input connections with progressively larger weight magnitude, thus gradually reducing the size of the decoding pool. However, given the very similar weight magnitudes of the component and pattern cells, the level of pruning did not alter their relative proportion in the decoding pool–this proportion remained the same as in the original population (i.e., 80% vs. 20%; Fig 5C, bottom). This confirmed the intuition (see Fig 5A) that training with gratings, even in presence of asymmetric cross-orientation suppression (i.e., affecting component but not pattern cells), leads to a decoding pool that simply reflects the proportion of component and pattern cells of the overall cortical population and, as such, is component-enriched.

A very different scenario emerged when we simulated the training of the P group in the plaid discrimination task (see Fig 5D). The weight distribution learned by the classifier for the pattern cell population (top, orange dots) was the same as in the grating discrimination task. This is because our simulated pattern cells were by definition immune to cross-orientation suppression. On the other hand, the weight distribution for the component cell population (green dots) was very different from that obtained in the grating discrimination task. First, since these cells responded to the components of the plaids (i.e., their constituent gratings), the classifier learned to assign larger weights to those units with preferred direction at ±60˚ (i.e., half plaid cross-angle) with respect to the global directions of the leftward- and rightward-drifting plaids (same as for the simulations shown in Fig 4C). In addition, because the

simulated components cells were affected by cross-orientation suppression, the classifier learned to rely less on component than on pattern cells, thus assigning lower weights to the former than to the latter (compare the absolute height of the peaks in the distributions of green and orange dots in Fig 5D, top). As a result, when the sparsity of the connections to the decision neuron was increased, by leaving in the decoding pool only inputs with progressively larger weight magnitude, the proportions of component and pattern cells in the pool followed opposite, mirror trends (Fig 5D, bottom). While the former decreased (green curve), the latter increased (orange curve), eventually leading to a reversal of the fraction of component and pattern cells in the decoding pool when the decision neuron was pruned of 90% of its potential inputs. This confirmed the intuition (see Fig 5B) that cross-orientation suppression can, at least in principle, strongly bias the recruitment of pattern cells in the decoding pool, under the constraint of sparse connectivity to the decision neurons, thus leading to a pattern-enriched representation despite the prevalence of component cells in the overall cortical population.

As a control experiment, we repeated the whole simulation without any cross-orientation suppression (S4 Fig). As expected, regardless of whether the classifier was trained with drifting gratings or plaids, it assigned similarly large weights to the component and pattern cells that better supported the discrimination task (S4 Fig, top). Thus, the only factor controlling the proportion of component and pattern cells in the decoding pool was the much larger number of the former in the overall population of available motion detectors. As a result, even when the sparsity of the connections to the decision neuron was progressively increased, the relative proportion of component and pattern cells in the decoding pool did not change. For both the simulated G and P groups, the component cells dominated the decoding pool (S4 Fig, bottom). Such component-enriched decoding pool, along with the lack of cross-orientation suppression, would produce similarly large identity- and cross-priming magnitudes for both groups, thus failing to account for the absence of cross-priming observed for the G group (Fig 3A). In addition, being the decoding pool mainly made of local motion detectors, the resulting identity- and cross-priming curves would follow different trends, as already shown in the simulations of Figs 4D and S3B.

### Simulating a scenario where a decision neuron reading out a pattern representation fails to be activated by plaids, if originally trained to discriminate drifting gratings

Despite the plausibility of the mechanism illustrated in the previous section, the key assumption of having weaker cross-orientation suppression on pattern cells than on component cells rests on hints from the primate literature only and has never been directly demonstrated in rodents. In addition, in our simulations, the decision neurons drew their inputs from a mixed population of pattern and component cells. That is, we assumed the decision neuron to pool from different stages of a putative motion processing hierarchy. Considering primates, this assumption is clearly at odd with the higher functional and anatomical rank of MT, with respect to V1, along the dorsal stream [3,4]. In monkeys, the MT representation would be the one to be read by downstream decision areas, no matter whether the incoming stimuli are gratings, plaids or even more complex patterns (e.g., hyperplaids or any other shape). Thus, decision neurons would not read out a mixed population of component and pattern cells (as we simulated in the previous section), as V1 component cells would only serve as a necessary, intermediate step to build up MT pattern cells. Decision neurons would only have access to the higher-order motion representation conveyed by MT pattern cells, regardless of the stimulus to be processed.

In the case of rodents, it is still unclear whether a similar hierarchy of dorsal areas exists, but at least one study has reported extra-striate area RL to be richer of pattern cells than V1

[16]. In addition, although an anatomical hierarchy of motion processing stages as structured as the one found in primates may not exist in rodents, pattern cells distributed over striate and extrastriate cortex could still sit above component cells along the functional motion processing chain. In fact, pattern cells, given their ability to encode motion direction in a shape-independent way, would behave as better motion detectors than component cells in virtually any visually guided behavior. As such, motion information could still be preferentially routed from visual cortical areas (e.g., V1, RL, LM) to higher-order decision centers (i.e., frontal and parietal cortices) mainly via pattern cells. This possibility is consistent with the well-established ability of distinct subpopulations, within mouse V1, to selectively make (receive) target-specific projections with (from) downstream (upstream) areas [56,67–70]. In addition, it is consistent with a very recent study showing that multiple visual areas (V1, LM and AL) are causally involved in coding stimulus orientation and contrast [71]–a demonstration that the representation of key visual features is distributed among neuronal populations within distinct visual areas.

In light of these considerations, an alternative scenario to interpret our findings is one where: i) decision neurons receive inputs mainly from pattern cells, regardless of the motion discrimination task to be performed (i.e., the decoding pool is, by "construction", pattern enriched); and ii) pattern cells are as sensitive to cross-orientation suppression as component cells are. The question is whether these assumptions are consistent with the difference in the magnitude of cross-priming produced by plaids on the rats of the G group (Fig 3A) and by gratings on the rats of the P group (Fig 3B). To account for this difference, we reasoned that pattern cells connected to a given decision neuron could fire less in response to plaids (because of cross-orientation suppression) than in response to gratings. This could lead the decision neuron to form stronger synaptic connections with its afferent pattern cells in the case of training with plaids than with gratings, under the assumption that some compensatory mechanisms actively keep the weighted input to the decision neuron around a target setpoint. Such compensatory process would be fully consistent with the well-known role played by homeostatic plasticity in cortex, where the overall synaptic strength scales as a function of the magnitude of the input, so as to maintain the average firing rate of cortical neurons within a given range [72–74].

To verify the plausibility of these intuitions, we simulated a downstream decision neuron reading out the representation conveyed by a population of pattern cells with strong cross-orientation suppression. As done in the previous section, we implemented the decision neuron as a logistic classifier with a regularization term on the L2 norm of its "synaptic" weights, so as to simulate the effect of homeostatic plasticity. The classifier was trained to discriminate either gratings or plaids drifting in opposite directions (i.e., 0˚ vs. 180˚ motion direction). As a result of the cross-orientation suppression, the L2 norm of the population vectors fed as inputs to the classifier was larger for the gratings than for the plaids (Fig 6A). This led the magnitude of the weights to become larger in the case of training with the plaids than with the gratings (Fig 6B), with the L2 norm of the weights' vector being more than twice as large in the former case (Fig 6C).

This had the "homeostatic" effect of nearly equalizing the magnitude of the weighted inputs to the classifier, when this was fed with the same kind of stimuli that were used for training. As shown in Fig 6D, the weighted inputs resulting from presenting drifting gratings to the classifier that was trained to discriminate grating motion direction (solid black curve) were very close to the weighted inputs resulting from presenting drifting plaids to the classifier that was trained to discriminate plaid motion direction (solid gray curve). This would explain the strong identity priming observed for the rats of both the G and P groups in our experiment (Fig 3, black curves). At the same time, the large weights' magnitude resulting from the

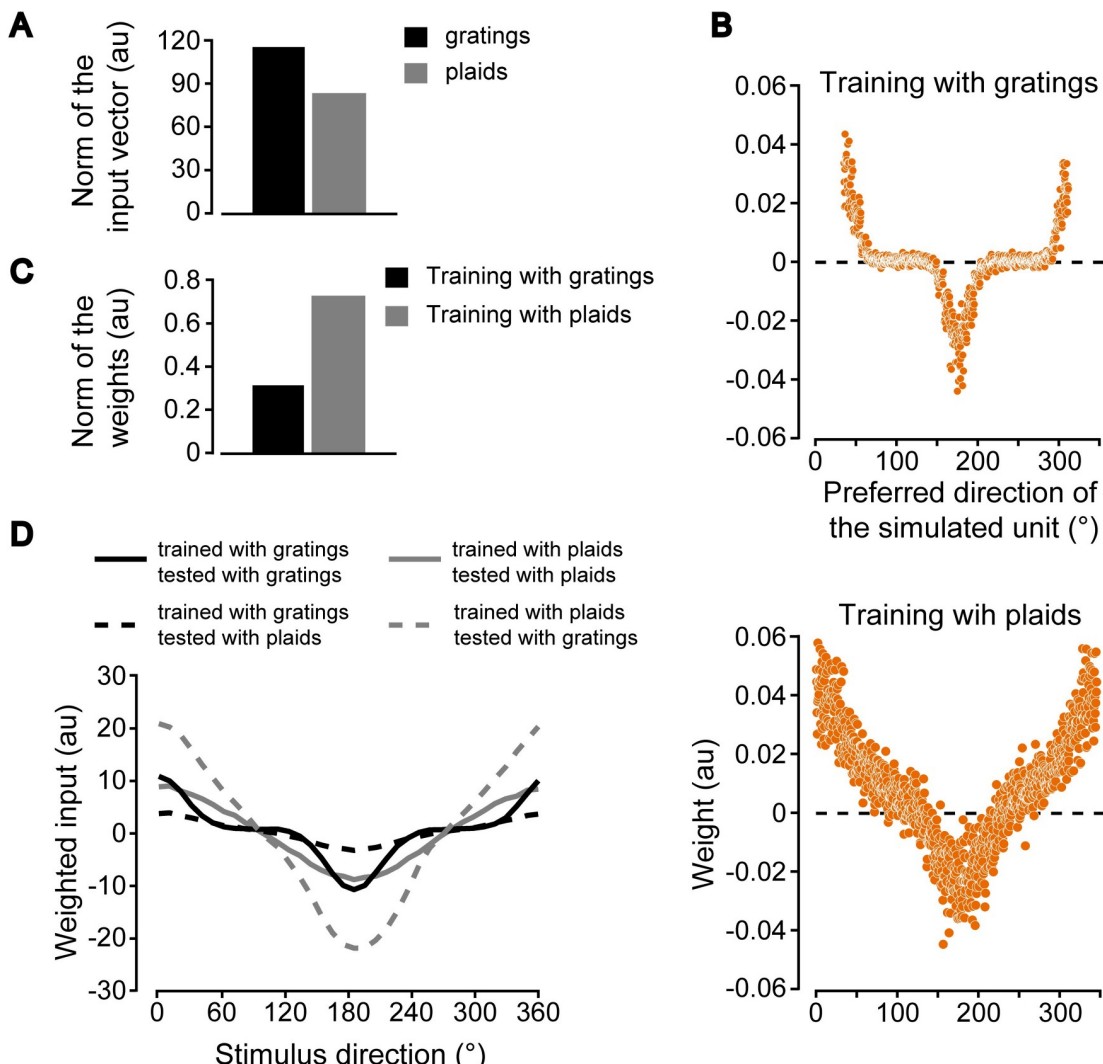

**Fig 6. Simulating a scenario where a decision neuron reading out a population of pattern cells fails to be strongly activated by drifting plaids, when originally trained to discriminate drifting gratings.** (**A**) L2 norms of the population vectors fed as inputs to a decision neuron (simulated as a logistic classifier with L2 regularization) that was trained to discriminate either a 0°-from 180°-drifting grating (black) or a 0°- from 180°-drifting plaid (gray). The classifier received inputs from a population of pattern cells. Each bar reports the mean of the norms of the input populations vectors produced by the 0°- and 180°-drifting stimuli. (**B**) Weights assigned by the logistic classifier to the population of pattern cells as a function of their preferred direction, in case of training with gratings (top) and plaids (bottom). (**C**) L2 norms of the weights shown in B. (**D**) Weighted inputs to the logistic classifier, originally trained to discriminate either gratings (black curves) or plaids (gray curves), when the population of simulated pattern cells was presented with drifting stimuli, homogeneously spanning the [0° 360°] range of directions. Such stimuli were either of the same kind (solid curves) or of a different kind (dashed curves) of those used during training. For instance, a classifier originally trained with plaids was tested with either plaids (gray, solid curve) or gratings (gray, dashed curve).

training with the plaids also guaranteed a very strong weighted input in the case of presentation of the grating stimuli (dashed gray curve in Fig 6D). This would explain the strong cross-priming observed for the rats of the P group (Fig 3B, gray curve). By contrast, the much smaller weights resulting from training with the gratings yielded a near-zero weighted input in the case of presentation of the plaid stimuli (dashed black curve in Fig 6D). This would explain the very small cross-priming observed for the rats of the G group (Fig 3A, gray curve). A plaid prime stimulus would simply be too weak (because of cross-orientation suppression) to activate the decision neuron through the weak, grating-adapted synaptic weights.

## Conclusions and implications

Although speculative in nature, the two explanations detailed in the previous paragraphs and backed up by our simulations propose an account for the behavioral data reported in our study. Importantly, they both rest on well-defined physiological assumptions. As such, they give rise to alternative hypotheses about the neural mechanisms underlying the processing of motion information that could be tested experimentally. For instance, the first explanation (Fig 5) would be supported by observing cross-orientation suppression acting way more strongly on component than on pattern cells in rodent visual cortex. On the other hand, the second explanation (Fig 6) would be corroborated by observing that cross-orientation suppression affects equally strongly pattern and component cells, and by finding a projection-specific enrichment in the proportion of pattern cells relaying visual information to decision making centers, such as posterior parietal cortex (PPC) [75–78].

In conclusion, our study not only yields behavioral evidence of motion integration in rodents, but also provides solid grounding for future investigations aimed at dissecting the neuronal circuits underlying integration of local motion cues into global motion percepts. In fact, as highlighted in a recent perspective [79], a behavioral task involving direction discrimination of gratings and plaids is a necessary ingredient of future experiments aimed at establishing a correlational and/or causal link between pattern-like responses observed in rodent visual cortex and integrated motion perception.

## Materials and methods

### Ethics statement

All animal procedures were conducted in accordance with the international and institutional standards for the care and use of animals in research and were approved by the Italian Ministry of Health and after consulting with a veterinarian (Project DGSAF 25271, submitted on December 1, 2014 and approved on September 4, 2015, approval 940/2015-PR).

### Animal procedures

We trained 21 male Long Evans rats (Charles River Laboratory) in a motion direction discrimination task. Upon arrival in the lab, rats weighed ~250 g, and they grew up to ~500 g. Animals started training during ~ 7th postnatal week. During the experimental period, they had free access to food, while their access to water was restricted. Their daily liquid intake included 5–15 ml of a 1:4 juice-water solution, plus an ad libitum water access for one hour after training. Training sessions lasted 50–70 minutes and took place 5 days per week.

### Behavioral apparatus and visual stimuli

The behavioral rig was the same previously used in several studies of rat visual perception carried out by our group [34,36–40]. It consisted of two racks, each equipped with three operant boxes to allow training a batch of six rats simultaneously [15,35]. In each box, one of the walls bore a 3 cm-diameter hole, so that a rat could extend its head outside the box and face frontally the stimulus display (a LCD monitor ASUS Ve228 located in front of the viewing hole at 30 cm). The box was equipped with three stainless-steel feeding needles (Cadence Science), located right outside the viewing hole, and serving as response ports and for reward delivery. To this end, each needle was connected to a led-photodiode pair to detect when the nose of the animal approached and touched it. The two ports positioned to the right and to the left of the hole were connected to two computer-controlled syringe pumps (New Era Pump Systems NE-500) for delivery of liquid reward, while the central port served exclusively to trigger the onset

of each behavioral trial. Each rat learnt to extend its head through the viewing hole, lick the central needle to trigger stimulus presentation and then lick either the left or right port to report the identity of the visual stimulus presented in the current trial. In case of correct response, the reward was delivered through the feeding needle. In case of an incorrect choice, a 1–3 s timeout period started, with the stimulus display flickering from black to middle gray at a rate of 15 Hz (during this period, a "failure" sound was also played). Stimulus presentation, response collection and reward delivery were controlled via workstations running the open source MWorks software (https://mworks.github.io/). Pictures and CAD drawings of the behavioral rig, showing the location of the animal with respect to the stimulus display and response ports, can be found in Fig 6 of ref. [15], supplementary figure 1 of ref. [36] and Fig 1 of ref. [40].

As explained in the Results, the rats were divided in two groups. The animals in the G group were trained to discriminate gratings drifting horizontally in opposite directions, i.e., 180˚ (leftward) vs. 0˚ (rightward). The rats of the P group were trained to discriminate two drifting plaids, whose global motion direction was, again, either 180˚ or 0˚. The grating stimuli consisted in full-field, full-contrast, sine wave drifting gratings with a temporal frequency of 2 Hz and a spatial frequency of 0.04 cycles/˚. The plaid stimuli were constructed by superimposition of two half-contrast gratings (again 0.04 cycles/˚ and 2 Hz) with a motion direction difference of 120˚ (i.e., 120˚ plaid cross-angle). In the experiments with the priming paradigm (see below), the drift direction of the prime stimuli (either gratings or plaids) was randomly sampled in each trial from 19 possible directions (i.e., from 0˚ to 180˚ in steps of 10; see Fig 1).

## Experimental design—Training phase

As soon as a rat licked the central response port, the target stimulus appeared (either a grating or a plaid depending on the animal's experimental group) and remained on screen for 2 s. To prevent rats from making very quick, impulsive responses, a trial was aborted if the animal's reaction time was lower than 300 ms. In such a case, the animal's response was not evaluated (neither reward or time-out was administered), the stimulus was immediately turned off, and a brief tone was played. Such "too fast" trials were also excluded from the analysis. Similarly, trials in which the rat responded more than 2 s after the offset of the stimulus were considered "ignored" and excluded from the analysis. No reward was delivered in these invalid trials. In order to accomplish the task and receive the reward, the rat had to reach and lick the response port matching the global motion direction of the target stimulus. Correct execution of such task was rewarded equally for both directions during each experimental session. In correct trials, the target stimulus remained on the screen for the whole reward delivery period, so as to strengthen the operant association between stimulus response and reward. On the other hand, when the animal provided an incorrect response, reward was not delivered, the stimulus was immediately turned off and the timeout period started (see previous section). The results of the training phase are shown in Fig 2.

## Experimental design—Priming paradigm

In the priming paradigm, a prime stimulus was shown to the animal before the target stimulus. The prime duration was 75 ms, the inter-stimulus interval (ISI, i.e., the interval separating the offset of the prime from the onset of the target) was also 75 ms and the target duration was 750 ms. As in the training phase, "too fast" or "ignored" trials were excluded from the analysis. Only, a trial was defined as "ignored" if the rat responded more than 1 s after the onset of the target stimulus. As a result, the allowed response window extended from 300 to 1000 ms from target onset. As in the training phase, an animal, in order to obtain the reward, had to reach and lick the response port (right or left) corresponding to the global motion direction of the

target. Accordingly, no feedback was ever provided to the rat regarding the motion direction of prime stimuli. In this paradigm, the target was not kept on screen during the reward, because there was no longer any need to favor the association between response, stimulus, and reward. Only animals maintaining a performance in the training task greater that 70% for 4 consecutive days were tested with this priming paradigm.

The rationale of this design rests on well-established findings in the human psychophysics literature [41,80–82], where presenting an "adapter" before a "test" stimulus has been shown to bias the perception of the test, depending on two key factors: the duration of the adapter and the inter stimulus interval (ISI) between adapter and target presentation. When the bias induced by the adapter attracts the perceptual choices towards the identity of the adapter itself, this effect is called "priming". Vice versa, when it repels the perceptual choices away from the identity of the adapter, it is named "adaptation after-effect". Our group has previously shown that brief presentation of a static shape (~50 ms), followed by a short ISI (66 ms), is able to induce a strong and robust priming effect on rat response to a target shape [34]. Here, we adapted this paradigm to moving stimuli, and we relied on previous motion priming studies in humans [41] to select the timing parameters of the task, in the attempt of inducing a strong priming effect.

### Analysis of behavioral data

Only rats displaying a strong identity-priming effect where included in the analysis described in the main text. Quantitatively, this corresponded to setting a threshold on the absolute priming magnitude in the identity priming condition amounting to 5%. Priming magnitude was computed averaging together the mean values of the absolute differences between the left and right extremes (4 points per side) of each priming curve with respect to the neutral prime condition (insets in Fig 3). The enforcement of such criterion led to the rejection of one rat per group, bringing the number of rats included in the analysis to 10 animals for the G group and to 9 animals for the P group out of the original 11 animals trained in grating direction discrimination and 10 trained in plaid direction discrimination.

To statistically compare the identity- and cross-priming curves obtained for each group of rats (black and gray curves in Fig 3) we performed a bootstrap analysis, in which the recorded behavioral sessions were resampled 50 times with replacement for each animal independently. This gave us the possibility to estimate a bootstrap standard deviation for both the average identity- and cross-priming curves of Fig 3, as well as for the average priming magnitudes computed as described above (insets in Fig 3). Bootstrap standard deviations were then used to compute normal 95% confidence intervals for the aforementioned quantities (displayed in Fig 3 as shaded regions around the priming curves and as error bars in the bar plot shown in the insets) [83]. In computing confidence intervals for the 19 points of a priming curve, the critical value corresponding to the chosen confidence level was adjusted for multiple comparisons using Bonferroni correction. A similar bootstrap procedure was used to obtain the confidence intervals in S1 Fig.

### Computational modeling

To interpret the results of our psychophysical tests, we simulated a decision neuron reading out the activity of a simulated population of either component or pattern cells fed with either grating or plaid stimuli. Three different simulations were performed to test the consistency of our findings with three possible scenarios concerning the tuning of rat visual neurons for motion direction and the choice of the decoding pool by the decision neuron. These simulations have been already described in the Results and in the legends of Figs 4–6. Below, we

explain some additional technical details about how we modeled the populations of simulated component and pattern cells and how we implemented the logistic classifier used to simulate the decision neuron.

**Simulations to check the consistency of the priming curves observed for the P group with a pattern-based representation of global motion direction.** To check whether the shape of the priming curves observed for the rats of the P group (Fig 3B) was more consistent with a component-like or a pattern-like representation, we trained a logistic classifier to solve the same motion discrimination task that was administered to the rats (i.e., 0˚- vs. 180˚-drifting plaids), based on the responses of a simulated population of either component of pattern cells (Fig 4). We then tested how the trained classifier would decode the full range of drifting plaids and gratings used to obtain, respectively, the identity- and cross-priming curves in Fig 3B. To simulate the tuning for local and global motion direction of the populations of component and pattern cells, we used 24 Von Mises functions [51,52] centered at equispaced angles along the circle (spaced by 15˚). The von Mises functions, which are the circular analogs of Gaussian functions, have a width that is controlled by the parameter $k$, which, in turn, is equivalent to the inverse of the variance of a Gaussian distribution [53]. The peak value of the functions (and thus of the tuning curves) was set to 1 for all the simulated units.

For a pattern cell, the simulated tuning curve was the same regardless of whether the input stimulus was a plaid or a grating stimulus (compare matching rows in the top and bottom panels of Fig 4B). That is, by construction, each simulated pattern cell responded to the global direction of the plaid stimuli. Conversely, for a component cell, the simulated tuning curve was defined by a single Von Mises function in case of presentation of the grating stimuli (Fig 4A, top), but was defined by the superimposition of two von Mises functions (with the peaks being 120˚ apart) in case of presentation of the plaid stimuli (Fig 4A, bottom). That is, by construction, each simulated component cell responded to the local directions of the constituent gratings of the plaid stimuli, rather than to the global direction of the plaid. For both the simulated pattern and component cells, the Von Mises functions were summed to a constant background activity. In addition, to emulate the variability of neuronal firing in response to identical stimuli, in each simulated trial, the actual response to a given stimulus was affected by Gaussian noise with zero mean and $\sigma = 0.25$. The $k$ parameter of the Von Mises functions of the simulated pattern and component cells was set to 7 in the main simulation shown in Fig 4, while it was allowed to range between 0.5 and 11 in the more extended simulations of S2 Fig. The ratio between the peak of the tuning curves and the background rate was set to 10.

To model the training received by the rats in our experiment, we sampled for 500 times the responses of the whole population of either pattern or component cells to each of the 0˚- and 180˚-drifting plaids used during the training phase. We then trained a logistic regression classifier to discriminate the two sets of 0˚ and 180˚ responses using gradient descent (implemented by the Matlab function "fminunc") and including and L2 regularization term in the cost function (whose parameter lambda was set to 1). The weights resulting from such training are those shown in Fig 4C and 4E. Finally, we fed to the classifier grating and plaid stimuli spanning the full range of possible global motion directions (from 0˚ to 345˚ in steps of 15˚) and we plotted the probability for each class of stimuli to be classified as moving rightward as a function of its direction (Fig 4D and 4F). The same simulations were also repeated by training the classifier with 0˚- and 180˚-drifting gratings (to simulate the results obtained with the G group) instead of 0˚- and 180˚-drifting plaids. The results of these simulations are shown in S3 Fig.

**Simulations to explore the mechanisms that could lead to a strong cross-priming only in case of training with plaid stimuli–scenario #1.** To obtain a proof of principle that the mechanisms illustrated in Fig 5A and 5B could lead to a decoding pool enriched of pattern

cells in case of training with plaids, we carried out a second simulation. As before, we trained a logistic regression classifier in the same motion direction discrimination task that was administered to our animals. This time, however, the response properties of the simulated component and pattern cells, as well as their relative proportion, better matched either the known properties derived from the rodent literature [16–18] or those inferred based on the monkey literature [20,21]. First, the simulated population of pattern and component cells was much larger (i.e., 1000 units). It was composed of Poisson firing neurons with either "component" or "pattern" tuning curves, built in the same way as described in the previous section. Second, their proportion was 80% and 20% respectively, so as to match the highest ratio of pattern to component cells observed in rodent visual cortex [16]. As in the previous simulation, we used Von Mises functions to define the tuning curves of component and pattern cells (see description in the previous section). In this case however, the preferred directions of the simulated neurons were not equispaced but drawn randomly from a uniform distribution over the circle. Baseline value of tuning curves was set to 2 spikes/second. Von Mises function of random amplitude (normally distributed around 8 spikes/s with standard deviation of 1 spike per second) were added to this baseline. The firing rate of the units in each trial was drawn from a Gaussian distribution with mean equal to the value set by the tuning curves and standard deviation of 2 spikes/s. The width (i.e., the parameter $k$; see previous paragraph) of the Von Mises functions was adjusted in such a way that the Orientation Selectivity Index (OSI) of the resulting population of component cells was distributed according to a Gaussian with a mean of 0.7 and a standard deviation of 0.1, while the Direction Selectivity Index (DSI) was distributed according to a Gaussian with a mean of 0.6 and a standard deviation of 0.2. OSI and DSI are standard measures of orientation and direction tuning, whose definition can be found, for instance, in [49,52,55,59].

To account for the imperfect cross-angle invariance documented by [17] we enlarged the width of Von Mises functions used to simulate the tuning of pattern cells to plaids by dividing the $k$ parameter previously used to define their tuning to gratings (see previous paragraph) by a factor 4. This made the responses of our simulated pattern cells better matched to the known properties of mouse pattern cells, reproducing their tendency to have broader tuning curves when tested with plaids than with gratings, especially when the plaids have a large cross-angle (such as the one used in our study, i.e. 120˚).

Finally, we simulated cross-orientation suppression in component cells–that is, we imposed that the peak responses of component cells to the plaids were half of their peak responses to the gratings, while pattern cells were implemented as being fully unaffected by cross-orientation suppression (i.e., same peak responses to plaids and gratings). This scenario was simulated because our goal was to test the hypothesis that, if cross-orientation suppression affected strongly component cells but not pattern cells, then the decision neuron would learn to preferentially connect with the latter, in the case of training with the plaid stimuli.

To model the training received by the rats in our experiment, we sampled for 200 times the responses of the mixed population of pattern or component cells to each of the 0˚- and 180˚-drifting stimuli (either grating or plaids) used during the training phase. We then trained a logistic regression classifier to discriminate the two sets of 0˚ and 180˚ responses using gradient descent (implemented by the Matlab function "fminunc") and including and L2 regularization term in the cost function (whose parameter lambda was set to 1). The weights resulting from such training are those shown in Fig 5C and 5D (top). Finally, we simulated a scenario where, as a result of this training, the decision neuron would maintain only the strongest of its synaptic connections with sensory neurons encoding the drifting stimuli. To incorporate this sparseness constraint on the connectivity of the decision neuron, we progressively pruned the weakest synaptic weights by applying a quantile-based thresholding to their magnitude after

training (i.e., by setting to zero any weight below the chosen quantile of the original weight magnitude distribution). We then quantified the proportion of pattern and component cells that survived the pruning and remained in the decoding pool of the decision neuron as a function of the quantile used as a threshold (Fig 5C and 5D, bottom). As a control, the whole simulation was also run without incorporating cross-orientation suppression in the responses of component cells (S4 Fig).

**Simulations to explore the mechanisms that could lead to a strong cross-priming only in case of training with plaid stimuli–scenario #2.** Here, we simulated an alternative scenario where a decision neuron (modeled, again, as a logistic classifier) is connected to a pool of sensory neurons made exclusively of pattern cells that are strongly affected by cross-orientation suppression. As such, the parameters of the simulations were the same as those described in the previous section, with only two notable differences: 1) no component cells were included in the decoding pool, but only pattern cells; 2) cross-orientation suppression affected pattern cells instead of components cells, and its impact was twice as strong as that of the previous stimulation (i.e., peak responses to plaids were 25% of peak responses to gratings). The training of the classifier was the same as described in the previous section. Note that the L2 regularization term in the cost function played the role of a homeostatic constraint on the overall weights' magnitude. Again, we compared the weights' distributions resulting from training with plaids and gratings (Fig 6B) and we also quantified the magnitude of weighted input (i.e., sensory "evidence") to the decision neuron, depending on the training and prime stimuli (Fig 6D).

## Supporting information

**S1 Fig. Rats of the Grating and Plaid groups display equally strong identity priming.** (**A**) Difference between the priming magnitudes measured for the G and P groups in the case of the identity-priming experiment (i.e., between the black bars in the insets of Fig 3A and 3B). Error bars are 95% confidence intervals obtained by bootstrap (see Materials and Methods). (**B**) Difference between the performances measured for the G and P groups with the neutral priming conditions, i.e., the 90°-drifitng (upward) graitng and plaid primes. Error bars are 95% confidence intervals obtained by bootstrap.
(TIF)

**S2 Fig. Simulating the discrimination accuracy afforded by a component-based representation with different degree of direction tuning.** (**A**) Example tuning curves over the direction axis, as obtained for increasingly large values of the parameter $k$, which controlled the width of the Von Mises functions used to simulate the tuning of the component cells (see main text). For each value of $k$, a population of 24 components cells was simulated, whose preferred directions homogeneously spanned the [0° 360°] range (same as in Fig 3A). The figure shows how one of such cells (the one with preferred direction = 160°) responded to drifting gratings (gray dashed curve) and plaids (black solid curve) as a function of their direction. For $k = 7$, the curves report the same values already shown in the rows of Fig 3A (top and bottom, respectively) corresponding to the preferred direction of 160°. (**B**) Weights assigned to the 24 simulated component cells by a logistic classifier that was trained to discriminate a 0°- (rightward) from a 180°-drifting (leftward) plaid. The parameter $k$ varies as indicated in A. (**C**) Fraction of times that the logistic classifier fed with the component-based representation (whose weights are shown in B) classified the test stimuli as drifting rightward (i.e., at 0°). The test stimuli were either plaids (solid black curve) or gratings (dashed gray curve) spanning the whole [0° 360°] range of directions. The parameter $k$ varies as indicated in A. (**D**) Absolute difference between the pairs of curves shown in C for plaid and gratings (averaged across all tested

directions) as a function of the parameter *k*.
(TIF)

**S3 Fig. Simulating the discrimination accuracy afforded by component- and a pattern-based representations for rats trained to discriminate drifting gratings.** Same simulations as those shown in Fig 4C, 4D, 4E and 4F, but the classifier was trained to discriminate 0˚- from 180˚-drifting gratings rather than 0˚- from 180˚-drifting plaids.
(TIF)

**S4 Fig. Simulating a scenario where training with both gratings and plaids leads to a decoding pool enriched of component cells.** Same simulations as those shown in Fig 5C and 5D, but without imposing cross-orientation suppression in the component cells.
(TIF)

**S1 Data. This zipped archive contains the source data (as a collection of .mat files) to produce all the plots presented in the main and supporting figures.**
(ZIP)

## Acknowledgments

We thank Daniele Bertolini for his help in developing and building the rigs for the behavioral experiments, and we thank Sofia Erica Rossi and Marco Salluzzo for their help in training rats in the motion discrimination task.

## Author Contributions

**Conceptualization:** Giulio Matteucci, Benedetta Zattera, Rosilari Bellacosa Marotti, Davide Zoccolan.

**Data curation:** Giulio Matteucci, Benedetta Zattera, Rosilari Bellacosa Marotti.

**Formal analysis:** Giulio Matteucci, Benedetta Zattera, Rosilari Bellacosa Marotti.

**Funding acquisition:** Davide Zoccolan.

**Investigation:** Giulio Matteucci, Benedetta Zattera, Rosilari Bellacosa Marotti.

**Methodology:** Giulio Matteucci, Benedetta Zattera, Rosilari Bellacosa Marotti, Davide Zoccolan.

**Project administration:** Davide Zoccolan.

**Resources:** Davide Zoccolan.

**Software:** Giulio Matteucci.

**Supervision:** Davide Zoccolan.

**Visualization:** Giulio Matteucci, Benedetta Zattera, Davide Zoccolan.

**Writing – original draft:** Giulio Matteucci, Benedetta Zattera, Davide Zoccolan.

**Writing – review & editing:** Giulio Matteucci, Benedetta Zattera, Davide Zoccolan.

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
