## [Decision Letter · Decision Letter 0]

22 Jun 2021

Dear Dr. Zoccolan,

Thank you very much for submitting your manuscript "Rats spontaneously perceive global motion direction of drifting plaids" for consideration at PLOS Computational Biology.

As with all papers reviewed by the journal, your manuscript was reviewed by members of the editorial board and by several independent reviewers. The reviewers appreciated your novel behavioral paradigm and its combination with simulations to yield testable predictions for future experiments. They raised important concerns however about the interpretation of your behavioral and modeling approaches. A notable concern is whether the priming differences could be due to the overall lower recognition performance of the plaid-trained rats. We believe follow-up experiments with matched stimuli or appropriate simulations are needed to clarify your findings. In light of the reviews (below this email), we would like to invite the resubmission of a significantly-revised version that takes into account the reviewers' comments. 

We cannot make any decision about publication until we have seen the revised manuscript and your response to the reviewers' comments. Your revised manuscript is also likely to be sent to reviewers for further evaluation.

Sincerely,

Leyla Isik

Associate Editor

PLOS Computational Biology

Thomas Serre

Deputy Editor

PLOS Computational Biology

Reviewer's Responses to Questions

**Comments to the Authors:**

Reviewer #1: The manuscript entitled ‚Rats spontaneously perceive global motion direction of drifting plaids‘ by Matteucci er al describes a very interesting test of pattern motion perception in rats. The authors trained rats to discriminate two opposite drifting directions of either a grating or in a different group of rats of a drifting plaid pattern consisting of two superimposed gratings.

The interesting point of this study was to present a brief priming stimulus before the actual test stimulation. This primer could be a grating or plaid drifting in any direction (out of a semi circle of 180 degrees).

By comparing the nature of the primer relative to the test stimulus (identical or cross) and the difference in angle of primer and target motion, the authors found a clear bias of the animals‘ decision by the direction of the primer. However, this bias was only found for rats trained with plaids. Those trained with gratings were also only biased by grating primers and not plaids.

The original aim of this study was to test if the populations of neurons representing gratings and plaids are independent of each other. The authors found both. In grating trained rats, the additional plaid primer had no effect. In plaid trained rats the additional grating primer however had the same strong bias effect as the identical plaid stimulus primer.

It is however unclear what the authors were trying to to prove. It is known from a large body of literature that pattern integration relies on component responses to gratings and pattern cells integrating these components into pattern responses. It is unclear how such an experiment should allow the dissection of these two populations.

It is in fact more obvious that the performance of the plaid trained rats is lower compared to those trained to discriminate gratings. Hence plaids are perceived less well. This might also explain why plaids have less of an effect as primers together with the better perceived gratings. It would have been clearer if the authors adjusted the stimuli to achieve similar performance, e.g. by using similar grey value distributions. Like this the results are very interesting but difficult to interpret.

In order to interpret their results the authors used different modeling approaches. First they simulated the discrimination accuracy afforded by component- and a pattern-based representations. Since component cells respond differently to gratings and plaids it is not surprising that populations of component cells also show different discrimination accuracy to these stimuli. And while pattern cell populations reliably show similar responses to gratings and plaids, they rely on input from component cells which’s not been included here.

The more interesting observation was that training the animals and their neurons with either gratings or plaids might boost their corresponding grating or plaid responding neurons. This might together with cross orientation suppression mechanisms be a likely scenario explaining the experimental results. Reverse training of the same rats in the study could have directly addressed this possible scenario even without additional neurophysiological recordings of neuronal tuning properties.

The second scenario is somewhat difficult to comprehend. The question is who tells the decision maker what drifting direction has been shown. The authors suggest „projection-

specific enrichment in the proportion of pattern cells relaying visual information to decision

making centers, such as posterior parietal cortex“. In this case one would expect pattern cells to be activated by plaid patterns should be better in guiding decision centers compared to activation by simple gratings.

Nevertheless the authors provide novel and interesting behavioral results that together with their simulation analysis provide predictions that can now be further tested experimentally. The manuscript however could be improved by clarifying the issues raised above.

Minor points:

The plaid pattern shown in figure one looks more like 60 degrees difference in the two components.

It would be helpful to show the behavior setup. Where was the screen relative to threat? We’re the stimuli showstopper the binocular field infringing or above the animals nose? This might also be a difference to the cited rodent studies since neurophysiology studies usually place the monitor to the side of the mouse or rat.

Reviewer #2: Matteucci et al. study motion perception in rats using behavior and computational model. Using a priming paradigm, they dissect whether rats rely on pattern motion or its components when trained to discriminate the direction of moving gratings or plaids. They find that rats trained on gratings are largely insensitive to the direction of motion of plaids priming stimuli, while those trained on plaids are primed by gratings moving in the same direction as the target plaids. Using simple models, they provide two alternative explanations of how training and the physiology of cortical neurons could result in the behavioral modulations by priming stimuli in the different groups.

While motion processing in rodents has been studied behaviorally before, the paradigm to identify the features the moving stimuli that rats learn to discriminate is very nicely designed and executed. The computational approach used to model the behavioral findings are rigorous and provide important insights. While inconclusive, as they result in two alternatives that remain untested, the models are carefully done, they rely on reasonable assumptions based on the literature that can be tested in future neurophysiological experiments.

I do not have major concerns with the approach and the conclusions. While the manuscript is in general well written and carefully explained, I found that it could be clearer in some sections . I also found some statements to be misleading or confusing:

1) The MS would benefit from a more direct comparison between the models in Fig 3 and Fig 5 and the behavior in Fig 2. While the priming effect on behavior in Fig 2 is plotted as a function of the difference between cross priming and target directions, in the models, the difference between grating or plaids is analyzed as a function of direction only. Comparisons would be clearer if the accuracy of the classifier or fraction of rightward choices were compared for the stimulus for which the classifier in the models was trained vs the “cross primed” stimulus as a function of their angle difference like in the behavior in Fig 2.

2) A model for the animals trained on gratings (G group) with no assumption of cross orientation or biased decision neurons is not directly discussed or included in Fig 3. While probably that model cannot explain the lack of cross priming effects of the G group (Fig 2A), it would be useful to discuss it to also get insights on the neural mechanism that the G group might be using to solve the task. If could also clarify why a different model for the G group might be needed to account for their small cross-priming effects as discussed in Fig 5.

3) The statement in lines 83-93 of the introduction, is confusing. “Critically….such as posterior parietal cortex (PPC) “ . While it is certainly an issue that animals might associate reward with a feature of the stimulus that is different from the one intended by the experimenter, in the case of RDK experiments it is not clear what these features would be. This stimulus, unlike plaids, is designed so that there are no local cues that are sufficient for inferring motion direction and that global motion can only be extracted from spatial pooling. What features of RDK could animals have been associating in previous behavioral paradigms that were different from the intented ones? In my view these previous reports are consistent with the P group in this study, in the sense that they show that rodents can learn to report motion using only global cues. As it is written it seems to state that ambiguities of the features that rodent learn using RDK needed to be clarified and are addressed in the MS.

4) Line 232 “ the neuronal population representing the direction of the gratings was largely inactive during the presentation of the plaids. “ . As no neuronal recordings were made in these experiments, this statement in the results sections is too speculative and misleading

**Have the authors made all data and (if applicable) computational code underlying the findings in their manuscript fully available?**

Reviewer #1: None

Reviewer #2: Yes

PLOS authors have the option to publish the peer review history of their article (what does this mean?). If published, this will include your full peer review and any attached files.

Reviewer #1: No

Reviewer #2: No
---

## [Decision Letter · Decision Letter 1]

1 Sep 2021

Dear Dr. Zoccolan,

We are pleased to inform you that your manuscript 'Rats spontaneously perceive global motion direction of drifting plaids' has been provisionally accepted for publication in PLOS Computational Biology.

Best regards,

Leyla Isik

Associate Editor

PLOS Computational Biology

Thomas Serre

Deputy Editor

PLOS Computational Biology

Reviewer's Responses to Questions

**Comments to the Authors:**

Reviewer #1: The authors have convincingly addressed my comments.

Reviewer #2: The revised version addresses all the issues I raised previously.

**Have the authors made all data and (if applicable) computational code underlying the findings in their manuscript fully available?**

Reviewer #1: None

Reviewer #2: Yes

PLOS authors have the option to publish the peer review history of their article (what does this mean?). If published, this will include your full peer review and any attached files.

Reviewer #1: No

Reviewer #2: No

---

## [Editor Report · Acceptance letter]

9 Sep 2021

PCOMPBIOL-D-21-00468R1 

Rats spontaneously perceive global motion direction of drifting plaids

Dear Dr Zoccolan,

I am pleased to inform you that your manuscript has been formally accepted for publication in PLOS Computational Biology. Your manuscript is now with our production department and you will be notified of the publication date in due course.

With kind regards,

Katalin Szabo
